# Quality of Life and Coping Strategies of Palestinian Women with Breast Cancer in the West Bank: A Cross-Sectional Study

**DOI:** 10.3390/healthcare13101124

**Published:** 2025-05-12

**Authors:** Ibtisam Titi, Nuha El Sharif

**Affiliations:** 1School of Public Health, Al-Quds University, Jerusalem 51000, Palestine; ibtisam.titi@students.alquds.edu; 2Ministry of Health, Ramallah P606, Palestine

**Keywords:** breast cancer, quality of life, coping strategies, Palestine

## Abstract

**Background/Objectives:** Breast cancer (BC) is the most prevalent cancer among Palestinian women and significantly affects their quality of life (QoL). Coping strategies are pivotal in shaping QoL outcomes; however, research examining coping strategies and QoL in the Palestinian context remains scarce. This study aims to evaluate coping strategies among newly diagnosed BC patients and their impact on QoL in the southern West Bank. **Methods:** A cross-sectional study recruited 147 newly diagnosed BC patients undergoing treatment in governmental hospitals in the Hebron and Bethlehem governorates. Data were collected via face-to-face questionnaires, which included the EORTC QLQ-C30, the Cancer Coping Questionnaire (CCQ), sociodemographic and clinical characteristics, and social support. **Results:** Participants exhibited moderate QoL scores, with physical functioning scoring highest (67) and emotional functioning lowest (49). Fatigue, insomnia, and pain were the most common symptoms. Coping strategies were moderately utilized, and global QoL was significantly associated with these strategies. Hierarchical regression showed education had a small positive effect on global QoL (R^2^ = 0.052, *p* = 0.005), while family support was a moderate predictor (R^2^ = 0.080, *p* = 0.041). The CCQ coping score negatively impacted global QoL (R^2^ = 0.186, *p* < 0.001), whereas CCQ positive focus (R^2^ = 0.342, *p* < 0.001) and diversion techniques (R^2^ = 0.406, *p* < 0.001) had substantial positive effects. **Conclusions:** Positive coping strategies, education, and family support play a vital role in enhancing QoL for newly diagnosed BC patients. Coping-focused interventions should be integrated into oncology care in Palestine to improve patient outcomes.

## 1. Introduction

Breast cancer (BC) is the most commonly diagnosed cancer among women globally, with approximately 2.3 million new cases reported in 2020. It also remains the leading cause of cancer-related deaths in females worldwide [1]. Although breast cancer is more prevalent in developed countries, over half of the cases occur in low- and middle-income countries, where it significantly contributes to the global cancer burden [2]. In Palestine, BC remains the most common cancer among women. In 2022, the Ministry of Health documented 934 new BC cases, with an incidence rate of 18.5 cases per 100,000 of the total population and 37.4 cases per 100,000 among women. Of these, 540 cases were reported in the West Bank, accounting for 15.8% of all newly registered cancer cases [3].

Quality of life (QoL), as defined by the World Health Organisation (WHO), refers to individuals’ perceptions of their position in life, which are influenced by cultural, social, and environmental settings, as well as their goals, expectations, standards, and concerns. This multidimensional concept encompasses physical health, psychological well-being, connection to others, and personal perspectives. It emphasizes subjective evaluations, focusing on how people perceive the influence of health problems and treatments on their overall well-being [4]. In oncology, improving patients’ QoL has become a primary goal of cancer treatment. QoL assessments have been shown to help predict treatment outcomes and prognosis [1,4]. Several studies have found that higher QoL measures are connected with longer survival rates for patients with various forms of cancer [5]. As a result, ongoing assessment and periodic evaluations of QoL and its predictors, from diagnosis through survivorship, are essential for improving patients’ quality of life and care management [6,7,8].

Quality of life varies among women newly diagnosed with breast cancer, those receiving different therapies, and long-term survivors who have completed treatment and returned to their regular lives [9]. Breast cancer is associated with reduced patients’ QoL [10,11]. The disease adversely affects multiple dimensions of well-being, including physical, psychological, and social functioning, ultimately reducing overall quality of life [12,13]. A combination of sociodemographic, medical, and psychosocial factors influences the QoL in BC patients [12,14,15].

Coping strategies are the active cognitive and behavioural responses individuals use to manage stressors. These strategies are generally classified into problem-focused coping, which directly addresses the stressor, and emotion-focused coping, which manages emotional distress [16,17]. Coping strategies significantly influence patients’ QoL and shape their perceptions of their illness [13,18]. Various coping strategies for managing cancer have been recognized, such as a fighting spirit, positive reframing, feelings of helplessness or hopelessness, and anxious preoccupation [12].

The effectiveness of these coping strategies can vary based on individual differences, the stage of cancer, treatment, and contextual factors that significantly impact the overall well-being of patients [19]. Early identification of coping strategies is crucial for managing distress and providing optimal patient care [20,21]. Additionally, effective coping strategies have been shown to positively impact treatment outcomes, survival rates, and the quality of life for women with breast cancer [12,22]. Newly diagnosed women with BC often utilize a variety of adaptive coping strategies, which are positively correlated with improved QoL [23]. However, the treatment protocols for these patients can negatively impact their QoL [24,25]. Therefore, to enhance health-related QoL, interventions must focus on promoting adaptive coping strategies and providing additional support to help patients adjust positively to their diagnosis [26].

Although BC is the leading cause of morbidity and mortality in Palestine, to our knowledge, there has been no adequate study done to assess QoL, its predicting factors, and coping strategies among BC patients. One study in Palestine used the Brief COPE scale and found that self-blame and religious coping were commonly used strategies among cancer patients experiencing death anxiety [27]. Understanding the relationship between coping strategies and QoL is essential for enhancing patient support and informing treatment decisions. Previously, we showed an urgent need to enhance the supportive care services for BC women within the Palestinian healthcare system, particularly in the areas of informational, physical, daily living, and psychological support care [28]. To our knowledge, this is the first study in Palestine that aims to evaluate the coping strategies adopted by newly diagnosed women with BC and their impact on their QoL in the Southern region of the West Bank.

## 2. Materials and Methods

### 2.1. Study Design

A cross-sectional design was employed to evaluate the association between QoL and coping strategies among newly diagnosed BC women in the southern region of the West Bank. This study design is efficient for exploring associations between variables and subgroups, making it ideal for public health surveillance, policy development, and quick, cost-effective research.

### 2.2. Study Context and Population

The health system in Palestine is comparable to those in other low-middle-income countries. The Palestinian Ministry of Health (MoH) serves as the primary healthcare provider, working in collaboration with non-governmental organizations (NGOs) and the private sector to promote cancer prevention and control [29]. All cancer cases in the southern region of the West Bank are referred to two governmental hospitals located in the Bethlehem governorate and the Hebron governorate. The two hospitals provide diagnosis, treatment, and inpatient services [30]. Despite these efforts, governmental hospitals face challenges in delivering comprehensive treatment due to limited financial resources and the sociopolitical and economic constraints that impact the availability and accessibility of cancer care [31]. These challenges are exacerbating the burden on individuals diagnosed with breast cancer [29,32].

The participants in this study were newly diagnosed women with BC receiving active cancer care at the oncology departments of governmental hospitals. According to the Division of Cancer Control and Population Sciences (DCCPS), newly diagnosed cancer patients are defined as individuals who have been diagnosed with cancer for the first time and are in active treatment, have completed active treatment, or are living with progressive symptoms of their disease [33]. Therefore, we used the term “newly diagnosed breast cancer” in this study context to refer to women who were receiving a breast cancer diagnosis within three months to one year from the time of inclusion in the study and are currently undergoing active treatment. Data were collected from BC women attending and receiving care at these hospitals between March and July 2024. Women who registered as new cases between August 2023 and July 2024 and were referred to the governmental hospitals in the southern region of the West Bank were invited to participate in this study.

### 2.3. Study Participants

According to the Palestinian Health Information Center (2023), a total of 2502 invasive breast cancer cases were registered in the West Bank between 2017 and 2021, of which 2465 were females, resulting in an annual average of approximately 500 cases. Notably, 21% of the diagnosed cases occurred in the Hebron governorate, while 9% were in the Bethlehem governorate [34]. In 2023, the Ministry of Health reported 540 newly diagnosed BC cases in the West Bank, of which 162 were from Bethlehem and Hebron governorates [3]. We aimed to include all newly diagnosed women with breast cancer (162 women) during the study period. However, we were able to reach and obtain consent from 147 women who agreed to participate. The women whom we were unable to contact were either undergoing therapy in the private sector, delaying their start of treatment, or explicitly refusing treatment.

### 2.4. Study Tools

Data were collected through face-to-face interviews with women receiving breast cancer care at governmental hospitals in the southern West Bank, using two questionnaires: the Arabic version of the European Organization for Research and Treatment of Cancer (EORTC) QLQ-C30 [35] and the Arabic version of the Cancer Coping Questionnaire (CCQ) [36].

#### 2.4.1. The EORTC QLQ-C30 Questionnaire

This includes 30 items divided into nine scales: five functional scales (physical, role, cognitive, emotional, and social), three symptom scales (fatigue, pain, and nausea/vomiting), and one global health and quality-of-life scale [35]. Each scale is scored from 0 to 100, with higher scores indicating better quality of life on the functional and global health scales, but greater symptom severity on the symptom scales [37]. The Arabic version of the EORTC QLQ-C30 used in this study has been validated for breast cancer patients [37,38]. The questionnaire demonstrated high reliability, with an overall Cronbach’s alpha of 0.90.

#### 2.4.2. The Cancer Coping Questionnaire (CCQ)

The CCQ has two scales: an individual scale (items 1–14) and an interpersonal scale (items 15–21). It has 21 items to evaluate coping strategies across cognitive, emotional, behavioural, and interpersonal domains. Each item is rated on a Likert scale from 0 to 4, with higher scores reflecting more frequent use of a particular coping strategy [36]. These scores provide insights into the frequency and effectiveness of the patient’s coping mechanisms.

In this study, CCQ was translated into Arabic and then back-translated into English to ensure linguistic accuracy. The tool underwent content validation by three experts in psychology and cancer care to confirm its relevance and clarity. To assess reliability, a pilot study was conducted with 30 breast cancer patients, and the internal consistency of the CCQ was evaluated using Cronbach’s alpha, which demonstrated high reliability (α = 0.94).

### 2.5. Statistical Analysis

All EORTC QLQ-C30 scale scores were linearly transformed to a 0–100 scale, following the formulas outlined in the EORTC QLQ-C30 scoring manual [39]. This transformation standardizes the scores, enabling comparisons across different quality-of-life domains. The global health status scale was used as the overall quality of life measure. A higher score represents better quality of life on the functioning scales and global health status, whereas on the symptom scales, it indicates greater severity of symptoms and poorer quality of life. In the analysis of the coping scale, two main components were identified: (1) the Total Individual Scale (Items 1–14), comprising four subscales—Coping (Items 2, 6, 7, 11, 12), Positive Focus (Items 1, 9, 14), Diversion (Items 3, 4, 8), and Planning (Items 5, 10, 13); and (2) the Interpersonal Scale (Items 15–21). The Arabic version of the Cancer Coping Questionnaire (CCQ) was employed after being validated by three experts.

Statistical analyses were performed using SPSS, version 25. Descriptive statistics summarized sociodemographic and clinical characteristics, with categorical variables reported as frequencies and percentages and continuous variables as means with standard deviations (SD). The normality of quality-of-life (QoL) data was assessed using the Kolmogorov–Smirnov test, with a significance level set at *p* < 0.05. Spearman’s rank correlation test was used to examine the strength and direction of associations between QoL and coping strategies, with statistical significance defined at *p* < 0.05. Bivariate analyses examined relationships between sociodemographic factors, clinical variables, and social support with QoL and coping strategies using the Mann–Whitney U and Kruskal–Wallis tests, as appropriate.

Before the analysis, all continuous variables of QoL and coping strategies were assessed for normality, outliers, and multicollinearity before being included in the linear regression model. The tolerance values ranged from 0.23 to 0.83, all exceeding the minimum acceptable threshold of 0.1. The Variance Inflation Factor (VIF) values ranged from 1.04 to 4.30, which are well below the critical value of 10, indicating that multicollinearity is not a concern. A weak association (r < 0.3) was observed between the QoL domains’ physical function, role function, all symptoms, total QoL, and coping strategies, except the global health status domain. Therefore, to investigate the impact of coping strategies on global health status (as the dependent variable), a stepwise hierarchical multiple regression analysis was conducted. The variables were entered in order, with each step enhancing the model’s explanatory power. Initially, sociodemographic factors (e.g., age, marital status, and education), medical variables (e.g., disease stage and time since diagnosis), and types of social support (e.g., family and spousal support) were included. Subsequently, coping strategies were added one at a time. The final model identified key predictors significantly influencing the global health QoL domain. We also applied Adjusted R^2^ to provide a more accurate measure of how well the model explains the variance in the dependent variable.

### 2.6. Ethical Considerations

We obtained approval from the Al-Quds University Research Ethical Committee (reference number REF. 10/24) and approval from the Health Education and Scientific Research Department of the Ministry of Health (reference number 162/475/2024). In addition, all participants were informed about the study objectives and signed an informed consent form emphasizing participant confidentiality and their right to withdraw from the study at any time without impacting their clinical treatments.

## 3. Results

### 3.1. Sociodemographic Characteristics of Study Participants

Table 1 presents the sociodemographic characteristics of the study participants. A significant proportion (71.4%) reside in the Hebron governorate. The participants’ mean age was 47 years (SD ± 10), with an age range of 25 to 70 years, and nearly half (49.0%) fell within the 41 to 54 age group. The majority of participants (87.1%) were married, 72.1% were housewives, 45.5% had attained a university or college degree, 68.0% had five or fewer children, and 63.9% lived in extended households consisting of more than five individuals. Of participants, 41.5% reported a monthly income ranging from USD 570 to USD 1140 (see Table 1).

### 3.2. Clinical Characteristics, Family History of Cancer, and Support in the Study Participants

Table 2 summarizes the clinical characteristics and supportive systems of the participants. Of the participants, 25.9% had been diagnosed within the past three months, 41.5% between three and six months, and 32.7% between six and twelve months. Half of the participants (45.6%) were diagnosed at stage 3, most of them (96.6%) underwent chemotherapy, 29.9% received radiation therapy, 9.5% received hormonal therapy, 19.0% underwent biological (immune therapy) treatment, and 59.2% had surgery. Among those who had surgery, 46.3% underwent a total mastectomy. Additionally, 33.3% of participants reported having chronic diseases other than breast cancer, and 59.9% used pain medications. Of the participants, 51.0% reported a family history of cancer, of whom 29.3% reported a history of breast cancer. BC patients reported receiving good social support (84.4%) from their families, but only 32.7% reported receiving support from their husbands (see Table 2).

### 3.3. Quality of Life Scale Analysis

Table 3 presents the descriptive statistics for the QoL scale. The transformed QoL scores provide detailed insights into breast cancer patients’ physical, emotional, and social well-being. The overall QoL score was moderate (score = 57). Among the functional scales, physical functioning received the highest transformed score of 67, while emotional functioning scored the lowest at a score of 49. For the symptom scales, fatigue emerged as the most prominent issue with a transformed score of 45, whereas nausea/vomiting (score = 19) was less frequent. These scores highlight the significant challenges patients face in managing their physical and emotional health, with fatigue being a prevalent issue, along with financial difficulties.

### 3.4. Cancer Coping Questionnaire (CCQ) Scales Analysis

The Cancer Coping Questionnaire (CCQ) scales provide valuable insights into the coping mechanisms employed by newly diagnosed women with breast cancer in this study. The highest mean score was observed in the CCQ diversion techniques scale (mean = 2.23, SD = 0.67), identifying it as the most frequently used method. This was followed by the CCQ positive focus scale (mean = 2.02, SD = 0.80), suggesting a moderate reliance on optimistic thinking. The CCQ planning scale (mean = 1.92, SD = 0.81) showed slightly lower engagement, reflecting some variability in forward-thinking strategies. The CCQ interpersonal scale had the lowest mean (mean = 1.61, SD = 0.76), indicating limited use of interpersonal relationships as a coping mechanism. The CCQ total individual scale (mean = 2.00, SD = 0.66) demonstrates moderate use of individual coping strategies, with noticeable differences among participants (see Table 4).

### 3.5. Correlations Between QoL and Coping Strategies

In this study, correlation coefficients (r) were below 0.3, indicating weak correlations between QoL domains and coping strategy variables, except for the global QoL health status that had better correlations with all coping strategies (r > 0.3) (see Table 5).

In Table 5, we observed significant negative correlations between coping strategies and multiple QoL domains. Physical functioning was negatively correlated with CCQ positive focus, CCQ planning, and the CCQ individual scale (*p*-value < 0.05). Role functioning also had negative correlations with CCQ positive focus, CCQ diversion, CCQ planning, and the CCQ individual scale (*p*-value < 0.05). In the domain of emotional functioning, CCQ diversion showed a significant negative correlation (r = −0.204, *p*-value 0.013) with QoL. Cognitive functioning was negatively associated with CCQ positive focus, CCQ diversion, CCQ planning, and the CCQ individual scale (*p*-value < 0.05). A similar correlation was found for social functioning, with significant negative correlations for CCQ positive focus, CCQ diversion, CCQ planning, and the CCQ individual scale (*p*-value < 0.05). Symptom items, including fatigue, pain, and dyspnoea, also had negative correlations with all these coping strategies, ranging from r = −0.169 to r = −0.266 (*p*-value < 0.05). CCQ positive focus, CCQ diversion techniques, CCQ planning, and the CCQ individual scales had negative correlations with financial difficulties (*p*-value = 0.001) (see Table 5).

Conversely, we found positive correlations between coping strategies and specific QoL dimensions. Emotional functioning was significantly positively correlated with CCQ coping (r = 0.180, *p*-value = 0.029). In the symptom domain, nausea and vomiting had positive correlations with coping, the CCQ individual scale, CCQ interpersonal factors, and CCQ planning, while there were weaker positive correlations with CCQ positive focus (*p*-value < 0.05). Additionally, CCQ coping was positively correlated with appetite loss (r = 0.203, *p*-value = 0.014). Global health QoL status demonstrated strong positive correlations with all coping strategies, including CCQ diversion, CCQ positive focus, and the CCQ individual scale (*p*-value < 0.001) (see Table 5).

### 3.6. Bivariate Analysis of Quality of Life, Coping Strategies, and Sociodemographic Variables

The analysis of sociodemographic variables and quality of Life (QoL) revealed several significant associations. Women living in refugee camps had higher mean ranks for emotional function, while urban women demonstrated higher mean ranks for global QoL health status (*p*-value < 0.05). Women aged ≥55 years reported higher mean ranks in physical and role functions (*p*-value < 0.05), whereas those aged ≤40 years had significantly higher mean ranks in emotional function levels (*p*-value < 0.001). Women with primary education or less had higher mean ranks in physical function and pain, while those with university education had better global health status scores (*p*-value < 0.05). Housewives also reported significantly higher mean ranks in physical function and pain scales (*p* < 0.05). Dyspnoea was more prevalent among women with more than five children (*p* = 0.03). Additionally, women with a monthly salary below USD 570 exhibited significantly higher mean ranks for physical functions, pain, insomnia, and financial difficulties (*p* < 0.05) (see Appendix A).

The analysis of coping strategies revealed significant associations between sociodemographic variables and the coping mechanisms used by participants. Urban women exhibited higher mean ranks for the CCQ coping and diversion techniques scales (*p*-value < 0.05). Women aged ≤40 years demonstrated higher mean ranks on the CCQ coping scales (*p*-value < 0.05). Married women also showed significantly higher mean ranks in positive focus and interpersonal coping strategies (*p*-value < 0.05). Education level played a crucial role, with women with higher educational attainment reporting significantly higher mean ranks across all coping strategy domains (*p* < 0.001). Additionally, housewives had higher mean ranks in the positive focus scale (*p* < 0.05) (see Appendix A).

### 3.7. Bivariate Analysis of Quality of Life, Coping Strategies, and Clinical and Social Support Variables

Clinical variables significantly influenced patients’ quality of life and coping strategies. Women diagnosed less than three months of treatment exhibited higher mean ranks for emotional functioning, nausea/vomiting, appetite loss, and constipation (*p* < 0.05). Those diagnosed with Stage 1 breast cancer had higher appetite mean ranks, while chemotherapy was associated with a lower mean rank in pain (*p* < 0.05). Surgical intervention was linked to lower mean ranks in emotional functioning, nausea/vomiting, insomnia, and appetite loss (*p* < 0.05). Women undergoing total or partial mastectomy, in particular, experienced lower scores in emotional functioning, nausea/vomiting, and appetite loss. Chronic diseases were associated with higher mean ranks in physical functioning (*p* < 0.05), and women using pain medication reported higher scores for fatigue, pain, dyspnoea, insomnia, and appetite loss (*p* < 0.05) (see Appendix A).

Regarding coping strategies, chronic diseases were the only clinical variable significantly associated, showing lower mean ranks in CCQ coping and CCQ planning scales (*p* < 0.05) (see Appendix A).

Social support also played a significant role. Family support was associated with a lower mean rank on the diversion coping strategy scale (*p* < 0.05). Husband support, however, was linked to a lower mean rank in physical functioning but a higher mean rank in emotional functioning and all coping strategy domains (*p* < 0.05) (see Appendix A).

### 3.8. Multivariate Analysis

A stepwise hierarchical multiple regression analysis identified key predictors of global quality of life (QoL) among participants. The model’s explanatory power increased cumulatively. Education was a significant predictor, with a small positive effect (B = 0.070, R^2^ = 0.052, *p* = 0.005). Family support also emerged as a moderate predictor, with higher levels of support associated with better global QoL (B = 0.249, R^2^ = 0.080, *p* = 0.041). In contrast, the CCQ coping score had a negative association with global QoL (B = −0.538, R^2^ = 0.186, *p* < 0.001), suggesting that maladaptive coping strategies reduce global QoL. Furthermore, both CCQ positive focus (B = 0.612, R^2^ = 0.342, *p* < 0.001) and CCQ diversion techniques (B = 0.689, R^2^ = 0.406, *p* < 0.001) had substantial positive effects on global QoL (see Table 6).

## 4. Discussion

This study aimed to evaluate the coping strategies adopted by newly diagnosed women with breast cancer and assess their impact on the quality of life in the southern region of the West Bank. The QoL analysis showed moderate functioning across physical, emotional, and social domains, with fatigue, insomnia, and financial difficulties being the most prevalent. Also, the study findings revealed significant associations between QoL and coping strategies. Positive coping strategies, like positive focus and diversion techniques, were linked to improved global QoL status, while maladaptive coping had a negative impact. Family support and education were also key factors in improving global health QoL, highlighting the importance of effective coping in managing breast cancer.

### 4.1. QoL Domains Assessment

Literature shows that BC significantly affects all dimensions of patients’ QoL [40]. An Australian study revealed that women newly diagnosed with breast cancer experienced significantly worse health-related quality of life on levels related to pain, physical functioning, health, and energy, compared to pre-diagnosis levels [41]. A study in Iran found that women with breast cancer reported significantly worse physical and role functioning, fatigue, and financial difficulties during the first six months compared to controls (*p* < 0.001) [42]. In this section, the study results of the QLQ-C30 domains will be discussed.

#### 4.1.1. The Global Health Status Domain

Patients were asked to rate their overall health and overall quality of life during the past week in this domain. This study’s global health status scores are considered moderate. This score is notably lower compared to findings from previous studies using the same measurement tool among BC patients. Most studies in the Arab countries that utilize the QLQ-C30 tool have reported varying global health status QoL scores. High QoL mean scores were observed in the United Arab Emirates, Saudi Arabia, Lebanon, and Jordan [43,44,45,46].

In contrast, moderate to poor QoL mean scores were reported in Tunisia, Morocco, Sudan, Egypt, and Kuwait [47,48,49,50,51]. In non-Arab countries, Ghanaian patients following mastectomy reported a considerably high QoL mean score of 83.3 [52], while Korean BC patients undergoing surgical interventions recorded an average score of 73.4 [53]. Similarly, Singaporean BC patients achieved a mean score of 74 [37], and Brazilian patients receiving radiotherapy without metastasis reported a score of 62 [54,55]. In contrast, the Ethiopian BC patients undergoing mastectomy had an average score of 48 [56], Iranian women receiving chemotherapy and radiotherapy reported a mean score of 50.5 [57]. Women with BC in Eastern China recorded an average score of 53.8 [58]. These disparities may reflect differences in healthcare systems, treatment approaches, and the availability of supportive care services among countries.

Studies conducted in Palestine on cancer patients using the EORTC QLQ-C30 have shown varying results. In the West Bank, a study assessing quality of life and post-traumatic stress disorder among adult females with cancer reported a global health status QoL score of 57.4 [59]. In the Gaza Strip, two studies among cancer patients receiving care in oncology departments at any stage documented global health status QoL scores of 49.9 and 58.9, respectively [60].

#### 4.1.2. Functional Scales Domain

Our results showed that among the five functional scales, physical functioning had the highest average score (67), while emotional functioning had the lowest (49). Our study’s scores were lower than those reported in other studies utilizing the EORTC QLQ-C30 questionnaire. For example, research in Korea reported scores of 83.8 for physical functioning and 83.3 for emotional functioning [53], while in Greece, the scores were 80.2 and 71, respectively [61]. In Bosnia and Herzegovina, a study reported scores of 66.32 for physical functioning and 36.58 for emotional functioning [40]. In Germany, the physical functioning was 68.9, and was 53.0 the emotional functioning [62]. Similarly, a study conducted in Brazil reported a physical functioning score of 81 and an emotional functioning score of 71.6 [63].

Studies from Arab countries reveal notable variations in physical and emotional functioning scores. Egypt reported the highest scores (78.4) for physical functioning and 68.6 for emotional functioning [64]. The United Arab Emirates scored 74.7 for physical and 68.4 for emotional functioning [46], while Saudi Arabia showed a distinct pattern with a physical functioning score of 62.14 and a higher emotional functioning score of 75 [44]. In contrast, lower scores were observed in countries like Kuwait, Morocco, and Jordan [46,47,50]. The diversity across countries may stem from the differences in healthcare systems, cultural norms, economic conditions, and patient characteristics.

The two local studies conducted in Gaza and the West Bank reported lower physical functioning scores compared to our study, with scores of 60 and 48.5, respectively. However, their emotional functioning scores were higher than ours, with scores of 64 and 77.8, respectively [59,65]. The differences between physical and emotional functioning in our study reflect the distinct challenges faced by BC patients. Previous QoL studies have identified BC treatment as a major factor impairing daily functioning, due to its physical and psychological impacts [66]. Higher physical functioning scores may suggest patients’ ability to perform daily activities with appropriate medical support, while lower emotional functioning scores emphasize the psychological burden of cancer, including fear, body image issues, and a lack of emotional support.

#### 4.1.3. The Symptom Scales Domain

Among the symptom scales, fatigue, insomnia, dyspnoea, and pain had the highest scores, indicating that these symptoms are the most prominent and significantly impact patients’ overall well-being. These findings align with previous studies, such as research conducted in Bosnia and Herzegovina, where fatigue, insomnia, and pain demonstrated higher mean scores [40]. Similarly, two Brazilian studies reported that pain and fatigue had the highest mean scores on the symptom scales [54,55]. Also, studies from Arab countries, including Saudi Arabia, Egypt, and Tunisia, identified fatigue, insomnia, and pain as the most prevalent symptoms [51,64,67]. Similarly, our findings align with local studies that reported fatigue, loss of appetite, insomnia, and pain as the most prominent symptoms [59,65]. These symptoms significantly impair QoL by affecting physical, emotional, and functional well-being. This multidimensional impact highlights the critical need for comprehensive symptom management strategies that address physical symptoms and should also provide psychological and social support.

### 4.2. Determinants of QoL Among BC

In this study, education was found to have a positive but modest association with QoL. This finding aligns with previous research demonstrating a correlation between higher education levels and improved QoL outcomes. For instance, Socha and Sobiech (2021) identified education levels as significant predictors of quality of life (QoL) among women after breast cancer treatment. Those with secondary education (12 years) or higher demonstrated better QoL outcomes compared to women with lower education levels [68]. Similarly, other studies have highlighted the role of sociodemographic factors, such as age, education level, marital status, and income, in influencing QoL. Specifically, higher education levels were associated with enhanced QoL, while lower education levels were linked to poorer outcomes QoL [69,70]. These findings suggest that higher education levels may enhance patients’ understanding of their disease, treatment options, and coping mechanisms, thereby contributing to improved overall quality of life.

Moreover, family support in this study exhibited a positive association with women’s QoL. The study findings align with existing literature that underscores the importance of emotional support from family in promoting well-being [71]. For instance, a longitudinal study conducted in India demonstrated that family support significantly enhanced global QoL and social functioning among BC patients, with these improvements becoming more pronounced over time. Such support was identified as vital in strengthening coping mechanisms and improving overall QoL [72]. Similarly, Mishra and Saranath (2019) highlighted the pivotal role of family support in addressing the socioemotional needs of BC patients, facilitating their adjustment to the diagnosis [73]. Additionally, A study conducted on 1160 women with newly diagnosed breast cancer in Shanghai, China, found that sufficient social support from family members, friends, and neighbours was significantly associated with improved quality of life in breast cancer patients [74].

In addition, the level of social support that a woman receives after being diagnosed with breast cancer has a significant impact on the types of coping strategies that she employs; social networks are an essential component in the management of the disease [20]. Furthermore, a study from Spain found that patients who perceived higher levels of support from family, friends, and the community reported better overall well-being and employed more effective coping strategies [75]. Family support is essential because it provides emotional, practical, and psychological resources that significantly improve well-being, particularly for individuals facing challenging health conditions like BC. This highlights the multifaceted role of family support in improving QoL. Moreover, in a 12-month prospective cohort study, Malaysian breast cancer patients were assessed at the time of diagnosis, as well as at 6 months and 12 months post-diagnosis. Findings indicated that perceived social support was significantly associated with improving quality of life and reducing emotional distress [76].

### 4.3. Coping Strategies Among Breast Cancer Patients

Coping mechanisms are crucial in the cancer survivorship journey, significantly impacting patients’ health. For women with breast cancer, coping strategies are essential for adapting to their illness. Factors like age at diagnosis, social support, and ethnicity are known to influence these mechanisms [77]. The most commonly used coping strategies among women with breast cancer include acceptance, emotional support, self-distraction, religion, active coping, planning, behavioral disengagement, and denial. These strategies significantly influence their adaptation to the diagnosis [78,79,80].

Our study findings reveal negative relationships between adaptive coping strategies, including positive focus, diversion, and QoL dimensions. A potential explanation is that too much dependence on positive focus or diversion may hinder patients from acknowledging the severity of their illness, leading to avoidance of essential medical follow-ups, self-care practices, or the processing of challenging emotions associated with the cancer experience. This could negatively impact their physical and emotional well-being over time. In addition, constantly trying to be positive or diverting attention might prevent patients from addressing genuine distress, pain, or unmet needs related to their diagnosis and treatment, and could result in social isolation and unmet emotional needs. This suppression can lead to an increased psychological burden and lower QoL [81,82].

For patients with newly diagnosed breast cancer, studies showed that the adoption of coping strategies varies. Some strategies, such as emotional support and acceptance, are positively associated with improved QoL, while others, such as denial and self-blame, are linked to poorer outcomes [23,83]. A study conducted in China found that newly diagnosed women frequently employ negative coping mechanisms, such as self-blame, rumination, and catastrophizing, which adversely affect their QoL. In contrast, strategies like acceptance, refocusing on planning, and positive reappraisal were found to positively influence QoL [84]. In our study, participants reported moderate coping strategy levels, with global QoL showing the strongest association with these strategies. Regression analysis revealed that the CCQ coping score negatively impacted global QoL, while CCQ positive focus and diversion techniques had substantial positive effects, emphasizing the critical role of coping strategies in improving global QoL.

Coping strategies that focus on positive reinterpretation and acceptance have been linked to higher QoL in breast cancer patients [17]. In Ethiopia, a study showed that approximately 51% of breast cancer patients used positive coping strategies [85]. On the other hand, maladaptive coping behaviors, such as avoidance, negative thinking, and emotional suppression, are consistently associated with poorer QoL, particularly among newly diagnosed breast cancer patients [25,86,87]. A study in South Korea highlighted that maladaptive coping in newly diagnosed breast cancer patients predicts poor health-related QoL. It emphasizes the need for regular screening of coping styles and interventions to improve maladaptive coping for at least two years after diagnosis [88]. Similarly, Gutiérrez-Hermoso et al. (2022) found that maladaptive strategies often correlate with increased symptoms of anxiety and depression, particularly at the onset of treatment. These psychological challenges can exacerbate distress, hinder treatment engagement, and ultimately reduce QoL [15].

Brunault et al. (2016) also noted that lower QoL in cancer patients is linked to frequent self-blame and limited use of positive reframing, underscoring the importance of adaptive coping strategies [89]. In line with this, Mohammadipour and Pidad (2021) emphasized that maladaptive coping can worsen psychological health, lead to treatment non-adherence, delay care-seeking, and accelerate disease progression, all of which further diminish QoL [90]. Therefore, coping strategies play a pivotal role in influencing QoL in breast cancer patients, with adaptive strategies enhancing well-being and maladaptive ones worsening outcomes. Integrating psychosocial interventions can foster effective coping and improve QoL.

Sociopolitical and economic constraints significantly influence the coping strategies adopted by breast cancer patients in the Palestinian context. The ongoing political instability, characterized by movement restrictions, checkpoints, and closures, severely limits access to essential healthcare services and specialized cancer treatments [32]. These barriers not only delay diagnosis and treatment but also contribute to psychological distress, thereby affecting how patients cope with their illness. Furthermore, financial difficulties exacerbate these challenges, as many critical supportive services, such as physiotherapy and psychological counselling, are not covered by governmental insurance, placing an additional burden on patients and their families [91]. Many of the coping strategies included in the Cancer Coping Questionnaire (CCQ)—such as planning for the future or managing daily routines—assume a level of personal stability and access to resources that these women often lack, making it difficult to engage in even basic coping efforts.

### 4.4. Association Between Coping and QoL of BC Patients

In this study, coping strategies emerged as significant predictors of quality of life (QoL), with the global health status scale being the only one strongly associated. While some strategies were negatively linked to QoL, reflecting the adverse effects of maladaptive behaviors, others demonstrated a beneficial influence. Our results showed that both the positive focus and diversion coping subscales were positively associated with global QoL, with positive focus contributing to 34% and diversion contributing to 41% of the variance in patients’ QoL.

These findings are consistent with the existing literature, which highlights the strong link between coping mechanisms and QoL in BC patients. For example, a study in Iran on breast cancer patients found that problem-focused coping strategies were significantly associated with higher total QoL, explaining 26.2% of the variance in the functional dimension and emerging as the strongest predictor of QOL among breast cancer patients [92]. Additionally, a Malaysian study found that effective coping strategies and positive behaviors significantly influence the quality of life in breast cancer patients [13]. Velasco et al. (2020) found that positive coping strategies emerged as the strongest predictor of the global QoL in breast cancer patients, accounting for 34.2% of the changes in QoL [93]. Another study in Turkey found that positive focus and distraction were significant predictors. Positive focus effectively reduced stress levels, while distraction slightly increased them, ultimately influencing the patients’ QoL [94]. These results suggest that engaging in adaptive coping mechanisms, such as focusing on positive aspects and engaging in distraction activities, can significantly enhance patients’ QoL.

## 5. Limitations

This study, while providing valuable insights, is subject to several limitations. Firstly, the study was conducted within the southern West Bank, potentially limiting the generalizability of the findings. However, given the homogeneity of the governmental healthcare system and cultural factors across the Palestinian territories, it is reasonable to anticipate that the observed trends may apply to a broader Palestinian population. Secondly, the cross-sectional design of the study precludes the establishment of definitive cause-and-effect relationships between coping strategies and QoL. Thirdly, the reliance on self-reported data may introduce inherent biases, as participants’ perceptions of their coping strategies and QoL may not accurately reflect objective outcomes. Fourthly, the study’s primary focus on women with breast cancer limits the generalizability of the findings to other cancer types and male patients. Fifthly, as the study exclusively included newly diagnosed patients, it fails to capture the long-term impact of coping strategies on QoL, necessitating future research employing longitudinal designs. Finally, logistical challenges, including financial constraints and political restrictions on movement, posed significant limitations to the study’s scope and execution.

## 6. Implications

This study underscores the critical role of effective coping mechanisms in enhancing QoL among individuals newly diagnosed with breast cancer. The use of adaptive coping strategies, including positive reframing and diversion techniques, is essential for effectively managing the multifaceted emotional and physical challenges associated with the disease. Furthermore, the literature indicates that comprehensive educational programs can significantly improve patients’ understanding of the disease, thereby promoting greater resilience and adaptability [95,96]. Acknowledging the crucial importance of family support, healthcare systems should actively engage family members in the care process through targeted training programs that equip them with the necessary skills to provide both emotional and practical assistance. Policymakers are encouraged to prioritize the integration of robust psychosocial support services within the framework of oncology care. This necessitates the establishment of multidisciplinary teams and the proactive reduction of cultural and socioeconomic barriers to ensure equal access to care and minimize the financial burden on patients. Future research should investigate the long-term impact of coping strategies on the QoL of breast cancer patients, with a particular emphasis on the development of culturally tailored interventions to optimize patient well-being and enhance their overall QoL.

## 7. Conclusions

This study provides valuable insights into the coping mechanisms employed by women newly diagnosed with breast cancer residing in the southern West Bank and their subsequent impact on QoL. The findings demonstrate that breast cancer significantly impacts all domains of QoL, with physical functioning exhibiting the highest scores and emotional functioning displaying the lowest. The presence of debilitating symptoms, including fatigue, insomnia, and pain, further exacerbates limitations in daily functioning and overall well-being. Coping strategies emerged as crucial determinants of QoL. The utilization of adaptive coping strategies, such as positive reframing and diversion techniques, was significantly associated with improved global health status and functional capacity. Conversely, maladaptive coping strategies were found to be correlated with poorer QoL outcomes. Furthermore, positive coping strategies, specifically positive reframing, and diversion, emerged as the most potent predictors of QoL, with education level and the availability of family support playing crucial roles in modulating these relationships. These findings underscore the critical significance of effective coping mechanisms in shaping the QoL of women newly diagnosed with breast cancer in this specific context.

## Figures and Tables

**Table 1 healthcare-13-01124-t001:** Sociodemographic characteristics of study participants (n = 147).

Variables	No.	%
**Patient governorate**		
Hebron	105	71.4%
Bethlehem	42	28.6%
**Place of residence**		
Refugee camps	11	7.5%
Village	92	62.6%
City	44	29.9%
**Age in years**		
≤40	40	27.2%
41–54	72	49.0%
≥55	35	23.8%
**Marital status**		
Unmarried	19	12.9%
Married	128	87.1%
**Education**		
Primary and less	20	13.6%
Secondary	60	40.8%
College/University	67	45.5%
**Working status**		
Employee	41	27.9%
Housewife	106	72.1%
**Number of children**		
≤5 persons	100	68.0%
>5 persons	47	32.0%
Family size		
≤5 persons	53	36.1%
>5 persons	94	63.9%
**Monthly salary (USD)**		
<570	54	36.7%
570–1140	61	41.5%
>1140	32	21.8%

**Table 2 healthcare-13-01124-t002:** Clinical characteristics and support for participants (n = 147).

Variables	No.	%
**Diagnosis duration (months)**		
Less than 3	38	25.9%
3–6	61	41.5%
6–12	48	32.7%
**Stage of disease at diagnosis**		
Stage 1	11	7.5%
Stage 2	58	39.5%
Stage 3	67	45.6%
Stage 4	11	7.5%
**Type of treatment**		
Chemotherapy		
Yes	142	96.6%
No	5	3.4%
Radiotherapy		
Yes	44	29.9%
No	103	70.1%
Hormonal therapy		
Yes	14	9.5%
No	133	90.5%
Biological (immune) therapy		
Yes	28	19.0%
No	119	81.0%
Surgical treatment		
Yes	87	59.2%
No	60	40.8%
Surgical intervention		
Total mastectomy	68	46.3%
Partial mastectomy	19	12.9%
No surgical intervention	60	40.8%
Yes	49	33.3%
No	98	66.7%
Taking pain medication		
Yes	88	59.9%
No	59	40.1%
Type of relative cancer		
Breast cancer	43	29.3%
Other cancers	32	21.7%
No cancer history	72	49.0%
Family support		
Yes	124	84.4%
No	23	15.6%
Husband support		
Yes	48	32.7%
No	99	67.3%
Other sources of support		
Yes	28	19.0%
No	119	81.0%

**Table 3 healthcare-13-01124-t003:** Descriptive analysis of EORTCQLQ-30C scales for women with breast cancer (n = 147).

Scales Items	Mean	Standard Deviation	Median	Range	Transformed Score (0–100)
**Overall QLQ-30 Score**	2.01	0.54	2.04	2.32	57
**Functional scales**					
Emotional functioning	2.54	0.78	2.5	3	49
Role functioning	2.26	0.77	2.00	3	58
Social function	2.22	0.91	2.00	3	59
Cognitive function	2.14	0.91	2.00	3	62
Physical functioning	2.00	0.73	2.00	3	67
**Symptom scales/items**					
Fatigue	2.36	0.65	2.33	3	45
Insomnia	2.31	1.11	2.00	3	44
Dyspnoea	2.22	1.09	2.00	3	41
Pain	2.21	0.69	2.00	3	40
Financial difficulties	2.14	1.01	2.00	3	38
Constipation	1.72	1.02	1.00	3	24
Appetite loss	1.71	0.91	1.00	3	23
Diarrhea	1.65	0.96	1.00	3	22
Nausea and vomiting	1.58	0.82	1.00	3	19
**Global health status**	3.72	1.08	3.50	6.00	45

**Table 4 healthcare-13-01124-t004:** Cancer Coping Questionnaire (CCQ) Scales Analysis of women with breast cancer.

Scales	Mean	Standard Deviation	Median	Range
CCQ-Coping	1.95	0.64	1.80	2.60
CCQ-Positive Focus	2.02	0.80	2.00	3.00
CCQ-Diversion Techniques	2.23	0.67	2.00	3.00
CCQ-Planning	1.92	0.81	2.00	3.00
CCQ-Total Individual scale (1–14)	2.00	0.66	2.00	2.86
CCQ-Interpersonal scale (15–21)	1.61	0.76	1.29	3.00

**Table 5 healthcare-13-01124-t005:** Spearman’s correlation, quality of life variables, and coping strategies.

QOL Variables	Coping	Positive Focus	Diversion	Planning	Individual Scale	Interpersonal Scale
Physical function	−0.102	**−0.162 ***	−0.111	−0.204 *	−0.163 *	−0.121
Role function	−0.086	**−0.179 ***	**−0.222 ****	**−0.202 ***	**−0.173 ***	−0.035
Emotional function	**0.180 ***	−0.039	**−0.204 ***	−0.036	−0.003	0.056
Cognitive function	−0.102	**−0.225 ****	**−0.189 ***	**−0.228 ****	**−0.197 ***	−0.120
Social function	−0.095	**−0.219 ****	**−0.265 ****	**−0.215 ****	**−0.197 ***	−0.006
Fatigue	−0.077	**−0.197 ***	**−0.256 ****	**−0.231 ****	**−0.192 ***	−0.015
Nausea and vomiting	**0.287 ****	**0.185 ***	0.071	**0.200 ***	**0.223 ****	0.202
Pain	−0.132	**−0.250 ****	**−0.236 ****	**−0.249 ****	**−0.227 ****	−0.070
Dyspnoea	**−0.188 ***	**−0.215 ***	**−0.169 ***	**−0.266 ****	**−0.235 ****	−0.158
Insomnia	0.093	−0.041	**−0.193 ***	−0.092	−0.031	−0.039
Appetite loss	**0.203 ***	0.097	−0.028	0.076	0.108	0.072
Constipation	−0.078	−0.102	−0.051	−0.101	−0.088	−0.003
Diarrhea	−0.012	−0.038	−0.073	−0.057	−0.037	0.012
Financial difficulties	**−0.211 ***	**−0.297 ****	**−0.288 ****	**−0.273 ****	**−0.275 ****	−0.123
Global health status	**0.407 ****	**0.573 ****	**0.597 ****	**0.500 ****	**0.545 ****	**0.318 ****

Bold indicates significant values (two-tailed) * *p*-value < 0.05, ** *p*-value < 0.01.

**Table 6 healthcare-13-01124-t006:** Multiple regression analyses predicting BC women’s QOL from sociodemographic, medical, social support variables, and coping strategies with the global health status scale (n = 147).

Variables	B	β	R^2^	F Change	*p*-Value/ Sig. F Change
(Constant)	1.537				
Education	0.070	0.046	0.052	8.032	0.005
Family support	0.249	0.084	0.080	4.237	0.041
CCQ Coping	−0.538	−0.319	0.186	18.797	0.000
CCQ Positive focus	0.612	0.458	0.342	33.589	0.000
CCQ Diversion	0.689	0.432	0.406	15.295	0.000

Method = Stepwise: Residence, Age, Marital status, Education, Work status, Children number, Family size, Income, Diagnosis duration, Stage of disease, Chemotherapy, Radiotherapy, Hormonal therapy, Biological therapy Surgical therapy, Surgery Type, Chronic diseases, Pain medication, Family History, Relative cancer types, Family support, Husband support, Other support. Method = Stepwise Coping 2. Method = Stepwise Positive Focus. Method = Stepwise Diversion.

## Data Availability

The original contributions presented in this study are included in the article/Appendix A. Further inquiries can be directed to the corresponding author.

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
