# Peer review of "Quality of Life and Coping Strategies of Palestinian Women with Breast Cancer in the West Bank: A Cross-Sectional Study"

_healthcare, 2025, doi:10.3390/healthcare13101124_

Round 1

Reviewer 1 Report

Comments and Suggestions for Authors
  • specify coping strategies that are considered adaptive or maladaptive in the Palestinian in the introduction
  • clarify the sampling process: purposive? convenience?
  • More information about psychometric validation results of the Arabic CCQ
  • Did you do  data triangulation and cross-validation for interviews.
  • it is good to have stratified analysis by disease stage or treatment modality
  • No discussion of religious/spiritual coping in discussion, can you add it 

Author Response

Dear Reviewer,

Thank you for your valuable feedback and for taking the time to review our manuscript. We truly appreciate your insightful comments and constructive suggestions, and we are grateful for your thoughtful review.

Please find below the responses to each point. Also, we incorporated the changes in red in the manuscript.

The authors

Comment 1: specify coping strategies that are considered adaptive or maladaptive in the Palestinian population in the introduction

Response 1: Thank you for pointing this out. We agree with your comment.

The CCQ scale we used in this study only measures adaptive strategies for breast cancer during treatment. we found only one study in Palestine, so we accordingly added this sentence in the introduction in line 81 page 2

“One study in Palestine used the Brief COPE scale and found that self-blame and religious coping were commonly used strategies among cancer patients experiencing death anxiety.” (Ahmead, Shehadah, and Abuiram 2024).

Comment 2 clarifies the sampling process: purposive? convenience?

Response 2: Thank you for pointing this out. We agree with your comment.

We did not do sampling.  We included all newly diagnosed women with breast cancer (162 women), but we could reach 147 women who agreed to participate. 

In line 130 page 3, we rewrote this paragraph to make it clearer to the readers:

 “We aimed to include all newly diagnosed women with breast cancer (162 women) during the study period. However, we were able to reach and obtain consent from 147 women who agreed to participate. The women whom we were unable to contact were either undergoing therapy in the private sector, delaying their start of treatment, or explicitly refusing treatment.”.

Comment 3 More information about the psychometric validation results of the Arabic CCQ

Response 3: Thank you for pointing this out. We agree with your comment.

We add this sentence, line 159 page 4

“In this study, CCQ was translated into Arabic and then back-translated into English to ensure linguistic accuracy. The tool underwent content validation by three experts in psychology and cancer care to confirm its relevance and clarity. To assess reliability, a pilot study was conducted with 30 breast cancer women, and the internal consistency of the CCQ was evaluated using Cronbach’s alpha, which demonstrated high reliability (α = 0.94).”

Comment 4. Did you do data triangulation and cross-validation for interviews.

Response 4: Thank you for pointing this out. We agree with your comment.

This study did not involve the collection of qualitative data or interviews, and it was not designed as a mixed-methods study; therefore, methodological triangulation did not apply here.

R2 scores presented in table 6 page 11 to enhance the validity of the regression model and assess its goodness-of-fit, which indicate the proportion of variance in the dependent variable explained by the model.

Comment 5 it is good to have stratified analysis by disease stage or treatment modality

Response 5: Thank you for pointing this out. We agree with your comment.

As presented in Supplementary Tables 5, 6, and 7, breast cancer stage and type of treatment were not significantly associated with any of the quality of life domains. Similarly, Supplementary Table 8 showed no significant associations between breast cancer stage or treatment type and any of the coping strategy domains.

Comment 6. No discussion of religious/spiritual coping in discussion, can you add it 

Response 6: Thank you for pointing this out. Unfortunately, the scale used in this study does not include items related to religious or spiritual coping strategies. However, this will be considered in future studies.

Reviewer 2 Report

Comments and Suggestions for Authors

Quality of Life and Coping Strategies of Palestinian Breast Cancer women in the West Bank: A cross-sectional study

This manuscript addresses a relevant and unexplored area by assessing quality of life and coping mechanisms among Palestinian women recently diagnosed with breast cancer. Comprehensive data collection, use of statistical analysis are the key strength of this study. The findings are useful for public healthcare particularly will help to deal with sociodemographic and resource challenges. Addressing the following points will enable more consistency, and a clearer interpretation before publishing this manuscript.

  • This study includes participants registered in the West Bank between 2017 and 2021 which required inclusion of more recent data for significant statistical interpretation with present scenario.
  • This study reveals negative relationships between adaptive coping strategies including positive focus, diversion and QoL dimensions. Explain these findings in detail with literature support by mentioning if they represent symptom overlap, reverse causality, or distress-driven coping.
  • Include a paragraph by stating how socio-political and economic constraints may shape the coping strategies used by participants.
  • Add clarity to the statistical regression model by including R², cumulative F-statistic for the final model. Also brief about multicollinearity checks.
  • The term “quality of life” is abbreviated differently throughout the manuscript like ‘QLQ’, ‘QOL’, and ‘QoL’, use one abbreviation across all sections.

Author Response

Dear Reviewer,

Thank you for your valuable feedback and for taking the time to review our manuscript. We truly appreciate your insightful comments and constructive suggestions, and we are grateful for your thoughtful review.

Please find below the responses to each point. Also, we incorporated the changes in red in the manuscript.

The authors

This manuscript addresses a relevant and unexplored area by assessing quality of life and coping mechanisms among Palestinian women recently diagnosed with breast cancer. Comprehensive data collection, use of statistical analysis are the key strength of this study. The findings are useful for public healthcare particularly will help to deal with sociodemographic and resource challenges. Addressing the following points will enable more consistency, and a clearer interpretation before publishing this manuscript.

Comment 1. This study includes participants registered in the West Bank between 2017 and 2021 which required inclusion of more recent data for significant statistical interpretation with present scenario.

Response 1: Thank you for pointing this out.

We did not include patients diagnosed outside the specified period; only women newly diagnosed with breast cancer in the year 2023 were included in the study (line 130 page 3).

Comment 2. This study reveals negative relationships between adaptive coping strategies, including positive focus, diversion, and QoL dimensions. Explain these findings in detail with literature support by mentioning if they represent symptom overlap, reverse causality, or distress-driven coping.

Response 2: Thank you for pointing this out. We agree with your comment.

There is a lack of studies that investigated the association between the CCQ scale and QoL. However, one of the possible explanations is that women who used the coping focuses and diversion strategies may not concentrate on their treatment, follow-up, and adherence that may increase the psychological burden and lower QoL.

According to your notes, we wrote these sentences in the manuscript to justify these results:  Line 497 page 14

“  A potential explanation is that too much dependence on positive focus or diversion may hinder patients from acknowledging the severity of their illness, leading to avoidance of essential medical follow-ups, self-care practices, or the processing of challenging emotions associated with the cancer experience. This could negatively impact their physical and emotional well-being over time. In addition, constantly trying to be positive or diverting attention might prevent patients from addressing genuine distress, pain, or unmet needs related to their diagnosis and treatment, and could result in social isolation and unmet emotional needs. This suppression can lead to an increased psychological burden and lower QoL (Gross & Levenson, 1997, Pennebaker, 1997).

Comment 3. Include a paragraph stating how socio-political and economic constraints may shape the coping strategies used by participants.

Response 3: Thank you for pointing this out. We agree with your comment.

Accordingly, we added this paragraph to clarify this association Line 541 page 15

“Socio-political and economic constraints significantly influence the coping strategies adopted by breast cancer patients in the Palestinian context. The ongoing political instability—characterized by movement restrictions, checkpoints, and closures—severely limits access to essential healthcare services and specialized cancer treatments (Rosenthal 2021). These barriers not only delay diagnosis and treatment but also contribute to psychological distress, thereby affecting how patients cope with their illness. Furthermore, financial difficulties exacerbate these challenges, as many critical supportive services—such as physiotherapy and psychological counselling—are not covered by governmental insurance, placing an additional burden on patients and their families (Abo Al-Shiekh et al. 2019). Many of the coping strategies included in the Cancer Coping Questionnaire (CCQ)—such as planning for the future or managing daily routines—assume a level of personal stability and access to resources that these women often lack, making it difficult to engage in even basic coping efforts.”

Comment 4. Add clarity to the statistical regression model by including R² and, cumulative F-statistic for the final model. Also brief about multicollinearity checks.

Response 4: Thank you for pointing this out. We agree with your comment.

R² and the cumulative F-statistic are added in Table 6 page 11.

For Multicollinearity we added the following description, line 189 page 5

“The tolerance values ranged from 0.23 to 0.83, all exceeding the minimum acceptable threshold of 0.1. The Variance Inflation Factor (VIF) values ranged from 1.04 to 4.30, which are well below the critical value of 10, indicating that multicollinearity is not a concern”. 

Comment 5. The term “quality of life” is abbreviated differently throughout the manuscript like ‘QLQ’, ‘QOL’, and ‘QoL’, use one abbreviation across all sections.

Response 5: Thank you for pointing this out. We agree with your comment.

Accordingly, we corrected the abbreviations 

Reviewer 3 Report

Comments and Suggestions for Authors

This cross-sectional study evaluated the impact on quality of life of newly diagnosis of breast cancer, the correlation between coping mechanisms and QoL outcomes, and the predictors of QoL, highlighted the role of effective coping mechanisms in enhancing QoL among individuals with newly diagnosed breast cancer. Overall, this manuscript is presented clearly and comprehensively.

Some minor points:

  1. line 41: Quality of life (QLQ) should be (QoL)
  2. line 102: The three hospitals should be two hospitals, correct?
  3. line 204-205: "68% had five or more children" whereas in Table 1, 68% ≤5 persons
  4. line 548: "comprehensive educational programs can significantly enhance patient understanding of the disease..." what is the proof behind this claim? there is no mention/evaluation of educational programs in the manuscript.

Author Response

Dear Reviewer,

Thank you for your valuable feedback and for taking the time to review our manuscript. We truly appreciate your insightful comments and constructive suggestions, and we are grateful for your thoughtful review.

Please find below the responses to each point. Also, we incorporated the changes in red in the manuscript.

The authors

This cross-sectional study evaluated the impact on quality of life of newly diagnosis of breast cancer, the correlation between coping mechanisms and QoL outcomes, and the predictors of QoL, highlighted the role of effective coping mechanisms in enhancing QoL among individuals with newly diagnosed breast cancer. Overall, this manuscript is presented clearly and comprehensively.

Some minor points:

Comment 1. line 41: Quality of life (QLQ) should be (QoL)

Response 1: Thank you for pointing this out. We agree with your comment.

Accordingly, we corrected the abbreviations  

Comment 2.  line 102: The three hospitals should be two hospitals, correct?

Response 2: Thank you for pointing this out. We agree with your comment.

 Accordingly, we corrected to two hospitals line  104 page 3

Comment 3.  line 204-205: "68% had five or more children" whereas in Table 1, 68% ≤5 persons

Response 3: Thank you for pointing this out. We agree with your comment.

Accordingly, we corrected to less that ≤5 persons line 217 page 5

Comment 4.  line 548: "comprehensive educational programs can significantly enhance patient understanding of the disease..." what is the proof behind this claim? there is no mention/evaluation of educational programs in the manuscript.

Response4: Thank you for pointing this out. We agree with your comment.

We fixed the sentence and added a reference Line 596   page 16

“Furthermore, literature showed that comprehensive educational programs can significantly enhance patient understanding of the disease, thereby fostering greater resilience and adaptability” (Correia et al. 2023; Sihvola, Kiwanuka, and Kvist 2023)

Round 2

Reviewer 2 Report

Comments and Suggestions for Authors

The manuscript may be accepted for publication.